# An Improved Protocol for *Agrobacterium*-Mediated Transformation in Subterranean Clover (*Trifolium subterraneum* L.)

**DOI:** 10.3390/ijms22084181

**Published:** 2021-04-17

**Authors:** Fernando Perez Rojo, Sumedha Seth, William Erskine, Parwinder Kaur

**Affiliations:** UWA School of Agriculture and Environment/Institute of Agriculture, The University of Western Australia, 35 Stirling Highway, Crawley, WA 6009, Australia; fernandoperezrojo@gmail.com (F.P.R.); sumedha.seth@uq.net.au (S.S.); william.erskine@uwa.edu.au (W.E.)

**Keywords:** subterranean clover, *Agrobacterium* transformation, in vitro culture

## Abstract

Subterranean clover (*Trifolium subterraneum*) is the most widely grown annual pasture legume in southern Australia. With the advent of advanced sequencing and genome editing technologies, a simple and efficient gene transfer protocol mediated by *Agrobacterium tumefaciens* was developed to overcome the hurdle of genetic manipulation in subterranean clover. In vitro tissue culture and *Agrobacterium* transformation play a central role in testing the link between specific genes and agronomic traits. In this paper, we investigate a variety of factors affecting the transformation in subterranean clover to increase the transformation efficiency. In vitro culture was optimised by including cefotaxime during seed sterilisation and testing the best antibiotic concentration to select recombinant explants. The concentrations for the combination of antibiotics obtained were as follows: 40 mg L^−1^ hygromycin, 100 mg L^−1^ kanamycin and 200 mg L^−1^ cefotaxime. Additionally, 200 mg L^−1^ cefotaxime increased shoot regeneration by two-fold. Different plant hormone combinations were tested to analyse the best rooting media. Roots were obtained in a medium supplemented with 1.2 µM IAA. Plasmid pH35 containing a hygromycin-resistant gene and *GUS* gene was inoculated into the explants with *Agrobacterium tumefaciens* strain AGL0 for transformation. Overall, the transformation efficiency was improved from the 1% previously reported to 5.2%, tested at explant level with Cefotaxime showing a positive effect on shooting regeneration. Other variables in addition to antibiotic and hormone combinations such as bacterial OD, time of infection and incubation temperature may be further tested to enhance the transformation even more. This improved transformation study presents an opportunity to increase the feeding value, persistence, and nutritive value of the key Australian pasture.

## 1. Introduction

Subterranean clover (*Trifolium subterraneum* L.) is the most widely grown annual pasture legume in southern Australia, with an estimated area of 29.3 Mha [1]. It is well-adapted to acidic soils, and there are subspecies adapted to neutral and alkaline soils, as well. It is mainly autogamous (inbreeding nature), diploid (2 n = 16), with a genome size of 540 Mb (cv. Daliak) divided into eight pseudo-molecules [2,3], and smaller than many other legume species [4]. In addition to the general benefits of legume pastures such as nitrogen fixation and soil structure improvement [4,5], it has three key additional features. Firstly, it is highly tolerant to intensive grazing as the growing points of its prostrate growth habit are close to the ground. Secondly, it buries its seeds, protecting them from predators and grazing and allowing reliable self-regeneration. Thirdly, it has a high nutritional value for livestock [6,7].

However, subterranean clover has limitations such as the need for relatively high available soil P levels due to its poor capability to explore the soil for P compared to other accompanying pasture grasses [8]. There is a huge scope for genetic improvement in subterranean clover to increase herbicide tolerance from inexpensive herbicide, reduce the capability of producing methanogenic compounds for rumens, overcome self-incompatible fertilisation, increase overall feeding value, persistence and nutritive value of the crop [4]. Therefore, it is required to optimise in vitro culture conditions for subterranean clover cv. Daliak. Only a few publications are available on the use of in vitro culture techniques to transform this species [9,10,11]. In all cases, direct organogenesis, using cut seeds as explants, was used to establish in vitro culture plants. Additionally, *Agrobacterium tumefaciens* transformation needs to be optimised as a delivery method in subterranean clover, as has been done for *Medicago truncatula* [12]. *A. tumefaciens* has largely become the method of choice for gene transfer in crop improvement [13,14,15]. In this project, we aimed to optimise the in vitro culture conditions and *Agrobacterium* transformation for subterranean clover cv. Daliak.

## 2. Results

### 2.1. In Vitro Culture Optimisation: Selection of Shooting and Rooting Media

The seeds, when grown according to the protocol of Ding et al. [10], showed contamination in 45.3 % of the cotyledons and in 57.7 % of embryo tips. However, the addition of 0.3% chlorine and 0.8 mg mL^−1^ cefotaxime in the seed sterilisation step reduced contamination—by comparison to 2%. In subsequent experiments, this percentage was maintained.

Regarding shoot regeneration, the medium RM73 worked appropriately for in vitro shoot regeneration with a frequency of explant regeneration of 80%. Additionally, it was necessary to identify which rooting medium was most suitable for subterranean clover cv. Daliak. Three out of the four rooting media tested did not show any root development after two months of culture. By contrast, as shown in Figure 1, explants cultivated into rooting media containing 1.2 µM IAA developed roots with 16.6 % efficiency after two months of culture.

### 2.2. Antibiotic Sensitivity Testing

#### 2.2.1. Kanamycin Concentration Test

Treatment with kanamycin at 100 mg L^−1^ reduced the percentage of explant survival to 23.3% after 30 days of culture (Table 1). Increasing the concentration of kanamycin in the solid media reduced explant survival (Appendix A). In some cases, all explant tissue died during the first days of exposure, whereas in others, death of the explant commenced with the yellowing of tissue followed by the cessation of explant growth.

As shown in Table 1, with 100 mg L^−1^ kanamycin, the survival rate was significantly different to other concentrations at 15, 30 and 45 days (*p*-value < 0.01). Additionally, it was also checked that there was no significant difference in the two independent experiments. Moreover, there was no significant difference between 100 mg L^−1^ at 30 and 45 days. A lethal dose of kanamycin could not be proved (no treatment killed 100% of explants).

Based on these outcomes, 30 days of treatment was selected with 100 mg L^−1^ kanamycin for the transformation experiments. This treatment could be expected to reduce explant survival to around 20%, and then a final selection using a PCR could be done to differentiate transgenic explants (kanamycin resistant) from false positives.

#### 2.2.2. Hygromycin Concentration Test

As previously described for kanamycin, hygromycin had a similar effect on explant survival (Table 1). Concentrations of 25 mg L^−1^ and 50 mg L^−1^ hygromycin showed a significant difference with the control (*p* value < 0.01) at 15, 30 and 45 days. Moreover, with 50 mg L^−1^ hygromycin a lethal effect was observed at 45 days. For that reason, it was decided to use 40 mg L^−1^ for subsequent experiments (50 mg L^−1^ hygromycin can have a negative effect on shoot regeneration of resistant explants, as well).

#### 2.2.3. Cefotaxime Concentration Test

The addition of cefotaxime to the medium did not reduce the percentage of survival over the control (0 mg L^−1^) at the concentrations studied (Table 1). Consequently, the addition of cefotaxime to the medium can be used to eliminate *Agrobacterium* without affecting the explants.

#### 2.2.4. Combined Antibiotics Test

As shown in Table 2, survival of explants treated with only 200 mg L^−1^ cefotaxime did not significantly differ from that of the control (0 mg L^−1^ kanamycin, 0 mg L^−1^ cefotaxime) confirming the above trial. In this experiment, the effect of 100 mg L^−1^ kanamycin was comparable with the previous one, but the percentage of survival decrease was greater (from 23% previous experiment to 13%). Additionally, the treatments containing kanamycin or hygromycin alone and its combination with cefotaxime did not show any significant differences. This result clarified that kanamycin or hygromycin can be combined with cefotaxime to select transgenic plants.

### 2.3. Cefotaxime Positive Effect on Explant Regeneration

A significant difference was observed in explant size when cefotaxime was added into the media (Appendix A). The number of shoots per explant was counted for each treatment (Appendix A). Considerable variability was detected within the treatments, including some outlier points. Nevertheless, a significant increase (*p* < 0.05) was found between the control (0 mg L^−1^) and the treatments 200 mg L^−1^ and 500 mg L^−1^ cefotaxime. As a result, cefotaxime significantly increased shoot regeneration in subterranean clover cv. Daliak from an average of 1.6 shoots per explant in the control to 3.4 shoots in 500 mg L^−1^ cefotaxime.

Looking at explant biomass, 200 mg L^−1^ cefotaxime increased (*p* < 0.05) the explant fresh weight from 25 mg (Control) to 77 mg (Appendix A).

### 2.4. GUS Plasmid

After *A. tumefaciens* strain AGL0 cultivation in a liquid medium, the plasmid was isolated, and a PCR was performed using primers specific to the GUS region. Three isolations were performed. In all cases, a clear amplification band of 331 bp was obtained, confirming the correct isolation protocol, as well as the correct plasmid (Appendix A).

### 2.5. GUS Transformation and Genetic Confirmation

Three independent experiments were completed. The number of resistant explants to hygromycin, as well as the number of positive PCR, was scored after 30 days of culture in selection medium (Appendix A). On average, 18.7% of the total explants survived the hygromycin treatment and 5.2% of these were confirmed as positives by PCR (*GUS* gene included into plant genome) (Table 3). From this result, around 13% of the total explants were resistant to hygromycin but not recombinant (false positives), confirming the results of the previous antibiotic combined test (15.5% of explant survival after 30 days in a medium with hygromycin plus cefotaxime). In all the experiments, the positive control (explants nontransformed in cefotaxime medium) showed a regeneration of around 80%, confirming previous results, and the negative control (explants nontransformed in cefotaxime plus hygromycin medium) showed a similar decrease in survival rate up to 10%, also confirming previous results.

## 3. Discussion

In this research, in vitro culture conditions, as well as the process of *Agrobacterium* transformation, were optimised for subterranean clover cv. Daliak, increasing the transformation efficiency from the previously reported 1% in other subterranean clover cultivars [10,11] to 5.2%.

To achieve this goal, firstly it was necessary to optimise the seed sterilisation protocol, due to a high contamination with fungi and bacteria that was found when a protocol from previous authors was applied [10,11]. Subterranean clover differs from all other clovers in its underground seed development (geocarpy) [1]. The geocarpic habit leads to a high microorganism load on the seed coat in the species, this being the main cause of contaminations and consequent death of explants. Totally contamination-free explants were not achieved in this study. The protocol to obtain subterranean clover explants was optimised using 0.3% chlorine plus 0.8 mg mL^−1^ cefotaxime. Modifications to these concentrations should be tested to eliminate contamination further. Testing of the addition of other antifungal compounds as published for other plant species [16] is also warranted. Additionally, the process of seed coat removal, embryo tip excision, and cotyledonary petiole cut is quite difficult to perform, even for a qualified technician. As some seeds were lost in the process, the final number of explants in every treatment differs, but that was taken into consideration when the statistical analysis was performed.

Regarding antibiotic sensitivity, 100 mg L^−1^ kanamycin reduced the survival of subterranean clover explants up to 23% in a chronic treatment at 30 days in line with other research with similar concentrations of this antibiotic in subterranean clover [10,11]. For hygromycin, as 50 mg L^−1^ produced a lethal effect, the concentration used for transformation experiments was 40 mg L^−1^. This hygromycin concentration was selected because concentrations above this level reduce transformation efficiency in other legume species [10,11], as well as for kanamycin [17,18]. In the case of cefotaxime, its addition into the media did not affect explant survival, making this antibiotic valuable to eliminate *Agrobacterium* after transformation without altering the explant growth.

Considering the combined use of antibiotics, kanamycin or hygromycin in combination with cefotaxime did not affect the effect of kanamycin or hygromycin *per se*. The utility of this combination of antibiotics in subterranean clover by previous researchers was confirmed [10,11].

Selected optimal antibiotic concentrations were used to optimise transformation efficiency. A transformation efficiency of 5.2% after one month of explant cultivation in selection medium was obtained using the plasmid pH35 and confirmed by PCR—higher than the 1% previously reported for subterranean clover [10,11]. Because there is a percentage of explants surviving the antibiotic selection (i.e., 15% of survival rate with 40 mg L^−1^ Hyg + 200 mg L^−1^ Cef) it is expected to have some selected but not transformed explants (less positivity rate with PCR than number of resistant explants).

Factors acting alone or in combination increasing this efficiency are as follows. (1) Due to the antibiotic sensitivity test, the most appropriate concentrations were used, and the percentage of escape explants (explants that survive antibiotic stress) was known, making the transformation more efficient. (2) The *Agrobacterium* strain AGL0 was used instead of the reported AGL1; perhaps AGL0 is more infective for subterranean clover explants (can be tested). (3) Using a direct PCR, it was unnecessary to extract DNA and then run the amplification, reducing error and increasing efficiency. Other variables may also be tested to further enhance the transformation efficiency, such as Agrobacterium OD and bacterial concentration (an OD = 0.4, 24/36 as incubation time was used according to previous reports [10,11]), time of infection, incubation temperature, concentration of acetosyringone, and co-cultivation period time.

Because transformation efficiency was tested at the explant level, it will be necessary to obtain a complete plant in further experimentation to test if the transgene is maintained through the generations in a Mendelian manner. There is also a need to test for chimeric explants, which can result in the development of both recombinant and nonrecombinant shoots within the same explant. This can be done using the *GUS* assay taking advantage that this gene was already inserted in the plant genome [19]. In addition, to check where the gene was inserted, as well as how many copies and its interaction with other genes, the whole plant genome can be sequenced using Hi-C technologies [20].

### Cefotaxime Positive Effect on Shoot Regeneration

It was demonstrated that cefotaxime produced a positive impact in explant regeneration and did not significative affect the survival rate (Table 1 and Table 2). Concentrations of 200/500 mg L^−1^ cefotaxime enhanced explant proliferation, increasing the number of shoots two-fold and their fresh weight three-fold, when compared with a negative control. This discovery was previously reported for other plant species [21,22,23], but this is the first report in subterranean clover. We suggest that future studies test 200 mg L^−1^ cefotaxime to improve subterranean clover in vitro culture efficiency and that this work be extended to other legume species. Thus, cefotaxime can play a dual role in the process of explant transformation—elimination of *Agrobacterium* following infection combined with its stimulus effect on shoot regeneration. The reason why cefotaxime acts as a positive growth regulator is not well understood. Some argue that cefotaxime mimics a plant growth factor [22], whereas others consider that degradation of cefotaxime by plant esterases might generate metabolites with growth regulation activity [21]. We also suggest that future studies should test the effect of cefotaxime on root formation. The efficiency of root formation using 1.2 µM IAA was found to be 16.6% [24], which might be improved with the addition of cefotaxime in the rooting medium. As an antagonist effect, 500 mg L^−1^ cefotaxime tended to reduce the mean explant fresh weight in relation to 200 mg L^−1^, but this did not reach significance (*p* < 0.05). In other plant species, such an inhibitory effect is caused by a high concentration of the antibiotic [25].

Considering all the solutions available to breeders to improve pastures cultivars from a molecular perspective, in vitro culture and *Agrobacterium* transformation, in combination with other technologies such as genome-wide association studies [26] and CRISPR [27], play a central role in the association of genes with agronomic traits. In this regard, analysing genes associated with biotic and abiotic stresses can unleash the power of plant adaptation, creating better pastures adapted to a range of different environmental conditions.

## 4. Materials and Methods

### 4.1. Seed Sterilisation

Seeds of subterranean clover cv. Daliak (sourced from Dr Phillip Nichols, (DPIRD) former Department of Agriculture and Food WA [DAFWA], 3 Baron-Hay Ct, South Perth, WA 6151, Australia) were washed for 3 min in 50 mL of sterile deionised (DI) water + 2 drops Tween−20 solution (Amresco^®^, Solon, OH, USA). Then, seeds were sterilised in 70% ethanol for 5 min. At this stage, 0.3% chlorine solution + 2 drops Tween 20 were added for 15 min. Subsequently, seeds were washed twice with DI water, and then moved into a 0.8 mg mL^−1^ cefotaxime (PhytoTechnology Laboratories^®^, Lenexa, KS, USA) solution for 10 min, modifying the protocol reported for Ding et al. [10]. Finally, seeds were washed 3 times with DI water (5 min each time) and left to imbibe overnight in a Petri dish with sterile water (Figure 2).

### 4.2. Explant Preparation and In Vitro Culture

The seed coat of each seed was removed, and then the embryo tip excised, leaving the cotyledonary explants with a 1–2 mm segment of cotyledonary petiole attached, following the protocol of Ding et al. [10] (Figure 3). The excision process was performed under a microscope (Leica MZ6^®^, Wetzlar, Germany) in sterile conditions. Then, explants were placed on a Petri dish (94 × 20 mm) containing 30 mL of Murashige and Skoog (MS) medium [28] supplemented with 30 g L^−1^ sucrose, 5 µM Thidiazuron (TDZ) (PhytoTechnology Laboratories^®^, Germany), and 0.5 µM Naphthaleneacetic acid (NAA) (PhytoTechnology Laboratories^®^, Germany) (hereafter RM73 regeneration medium [29]). Subcultures were made every 2 weeks onto fresh RM73 medium. The presence of contamination (fungal or bacterial) was verified at the end of week one. Once shoots were regenerated by direct organogenesis, they were individually excised and transferred onto a rooting induction medium.

Four rooting media were tested with different combinations of hormones, all of them containing half-strength MS salts and MS vitamins plus 15 g L^−1^ of sucrose: (i) 1.2 µM indole-3-butyric acid (IBA) [10] (PhytoTechnology Laboratories^®^, Germany), (ii) 3 mg L^−1^ IBA [11], (iii) 1.2 µM indole-3-acetic acid (IAA) (Sigma-Aldrich^®^, Munich, Germany) [24], and (iv) 5 µM IAA and 1 µM Kinetin (Sigma-Aldrich^®^, USA) [30]. The presence of roots was scored after 2 months of culture in rooting medium. In all media, the pH was adjusted to 5.75, and 8 g L^−1^ of agar (PhytoTechnology Laboratories^®^, Germany) was added for solidification, and explants were maintained at 22 °C with a photoperiod of 16 h light/8 h dark.

### 4.3. Antibiotic Concentration Test

Firstly, 100 seeds were used to produce explants as described above and these were cultured for 3 days on RM73 medium. Then, explants were subcultured onto Petri dishes containing 30 mL of RM73 medium plus various concentrations (0, 25, 50, 75, 100 mg L^−1^) of kanamycin (PhytoTechnology Laboratories^®^, Germany). Explants were maintained at 22 °C with a photoperiod of 16 h light/8 h dark and subcultures were made every two weeks as described previously. After 15, 30 and 45 days, the number of surviving explants was scored according to Figure 3. The experiment was repeated twice with kanamycin. The same experiment was performed for hygromycin, testing 0, 10, 25, and 50 mg L^−1^ of this antibiotic and repeated twice.

During transformation with *Agrobacterium tumefaciens*, another antibiotic is often used to eliminate the presence of this bacteria. In this case, the antibiotic was cefotaxime (PhytoTechnology Laboratories^®^, Germany). Because hygromycin or kanamycin are used concurrently with cefotaxime, the combined effect of these antibiotics was analysed and compared with a control (0 mg L^−1^ kanamycin, 0 mg L^−1^ hygromycin and 0 mg L^−1^ cefotaxime). In this case, the number of surviving explants was also counted, starting with 100 seeds and following the protocols previously described.

### 4.4. Cefotaxime Effect on Shoot Regeneration

In this case, 100 seeds were cut, and the resulting explants were cultivated following the above methodology. After 3 days of culture in RM73 media, explants were transferred into RM73 media supplemented with 100, 200 and 500 mg L^−1^ of cefotaxime (0 mg L^−1^ of cefotaxime was used as control). The number of surviving explants was counted after 15 and 30 days. Additionally, the number of shoots per explant, as well as their fresh weight, was recorded. For shoot count, explants were placed under the stereomicroscope (Leica MZ6^®^, Germany) in sterile conditions and scored according to Figure 3. In those cases where the shoots were not clearly identified, the whole explant was dissected, and shoots were counted separately. Individual shoots were considered when they were longer than 5 mm after 30 days in culture. Fresh weight was measured by transferring the explants onto a scale (Denver Instrument XL-610^®^, USA) in sterile conditions.

### 4.5. Plasmids and Bacterial Strains

*GUS* Plasmid (pH35)

*A. tumefaciens* strain AGL0 carrying the pH35 vector [17] was used to transform plant material. The plasmid components are shown in Appendix A. A polymerase chain reaction (PCR) with specific primers (F: ACAAGCAGAAGAACGGCATCA, R: CGATCCAGACTGAATGCCCA) was performed to amplify a segment of the *GUS* gene and check that the correct plasmid was inserted into *Agrobacterium*.

### 4.6. In Vitro Plant Transformation and Regeneration

Isolated explants were obtained, as described above, leaving a 1–2 mm segment of cotyledonary petiole attached. The *A. tumefaciens* strain AGL0 carrying the *GUS* plasmid (pH35) was cultivated 24/36 h (OD = 0.4) in Luria Bertani medium (LB, Sigma-Aldrich^®^) supplemented with streptomycin 50 mg L^−1^ and spectinomycin 100 mg L^−1^ (*GUS* plasmid selection, Sigma-Aldrich^®^, USA), plus acetosyringone 100 µM (Sigma-Aldrich^®^, USA). Explants were then placed in contact with *Agrobacterium* for 40 min in the dark in a 50 mL falcon tube containing 5 mL of bacteria culture. After this process, explants were transferred into a sterile filter paper and co-cultivated for 3 days on RM73 medium, following the protocol of Ding et al. [10]. Then, explants were rinsed with sterile water to remove the *Agrobacterium* excess, then incubated in cefotaxime 600 mg L^−1^ for 30 min to be then transferred onto a RM73 selection medium supplemented with 40 mg L^−1^ Hyg (for recombinant explant selection) and 200 mg L^−1^ cefotaxime (for *Agrobacterium* elimination). Two controls were used, placed in the same conditions as the treatment (in contact with LB media without *Agrobacterium* and imbibed with cefotaxime): a negative control (RM73 media with cefotaxime + Hyg) where most explants were expected to die; and a positive control (RM73 media with cefotaxime) where most explants were expected to remain alive. Subculturing was done onto a fresh selection medium every 2 weeks. The resulting antibiotic-resistant shoots were excised and subcultured into rooting media for root induction. The experiment was repeated 3 times, using 100 seeds (around 150 explants) each time.

### 4.7. Genetic Confirmation Test

To test the presence of the *GUS* gene in the plant genome, a direct PCR was conducted using the Phire^®^ Plant Direct PCR Kit (Thermo Scientific^®^, Waltham, MA, USA). The cluster of shoots obtained after 30 days of culture was scratched with a P20 tip and left in contact with the master-mix PCR solution for 1 min. The thermal condition selected was 5 min at 98 °C, followed by 40 cycles of 10 s at 98 °C, 10 s at 65 °C and 20 s at 72 °C, with a final extension step of 60 s at 72 °C (Primer sequences F: ACAAGCAGAAGAACGGCATCA, R: CGATCCAGACTGAATGCCCA), using the thermo-cycler SimpliAmp (Thermo Fisher Scientific^®^, USA). After the reaction was finished, an agarose gel electrophoresis was conducted, using a 100 bp ladder to identify the correct fragment size (331 bp). Transformation efficiency was calculated as the rate of positive amplifications divided by the total amount of explants used per experiment.

### 4.8. Statistical Analysis

A proportion test was performed to analyse differences in survival rate among the different antibiotic treatments. The null hypothesis argued that there were no significant differences between the populations. If the null hypothesis was rejected by a significant difference (*p* < 0.05), an individual set of proportion tests was performed comparing between every population.

Additionally, to test differences in explant weight and number of shoots, a one-way ANOVA was performed, argued with the same hypothesis as before. In this case, if the null hypothesis was rejected by a significant difference (*p* < 0.05), a Tukey test was implemented to analyse individual differences between the populations. In most cases, the data were unbalanced, indicating that the number of explants in each treatment was different. The Tukey test was performed considering a matrix for unbalanced data. For the statistical analysis, as well as for graphics generation, the software used was R Studio© (Version 1.0.153).

The various reagents used to carry out the experiments have been summarised in Appendix A.

## 5. Conclusions

Transformation efficiency was improved from the 1% previously reported in subterranean clover [10,11] to 5.2 %, tested at the explant level. Additionally, the finding of cefotaxime as a shoot regeneration enhancer in subterranean clover can be used to implement this antibiotic in combination with other plant hormones to increase explant regeneration and the number of shoots per explant starting from the same number of seeds. Moreover, we suggest that future research is warranted to test whether cefotaxime can be also used to increase rooting efficiency under in vitro conditions.

## Figures and Tables

**Figure 1 ijms-22-04181-f001:**
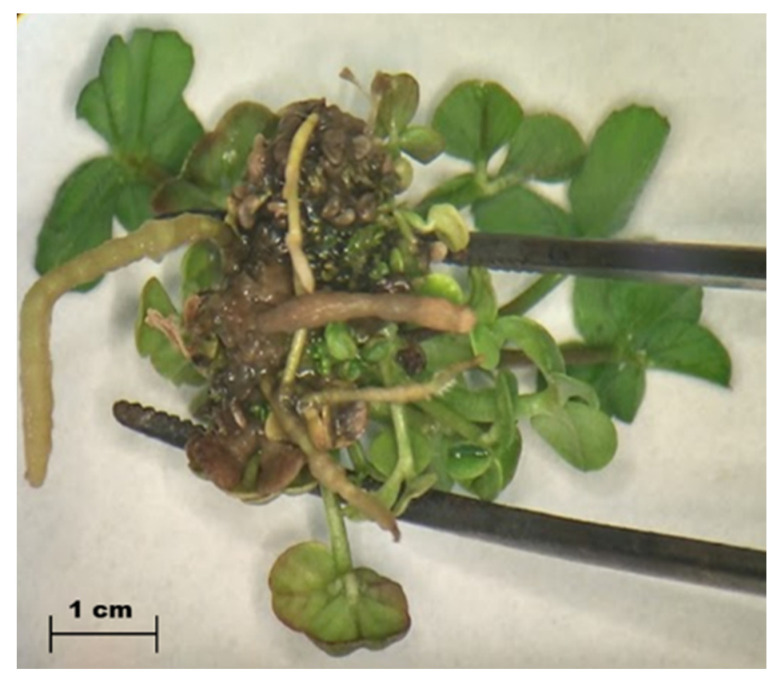
Explant cultivated for two months in rooting medium containing 1.2 µM IAA. Presence of multiple roots coming out from the bottom part of the explant. Image captured under stereo-microscope Leica MZ6^®^.

**Figure 2 ijms-22-04181-f002:**
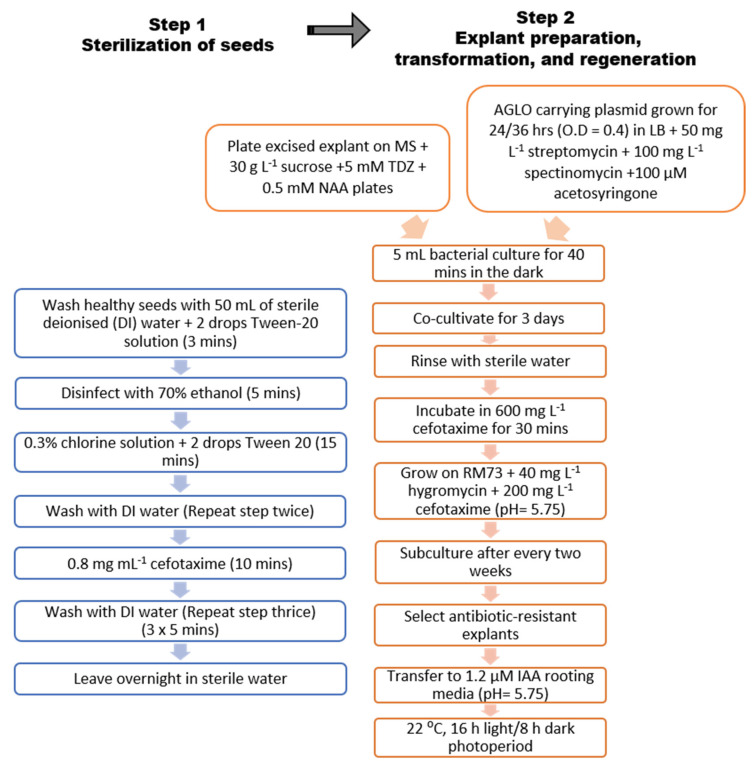
Schematic flowchart of the protocol including sterilisation of seeds and explant preparation, transformation, and regeneration.

**Figure 3 ijms-22-04181-f003:**
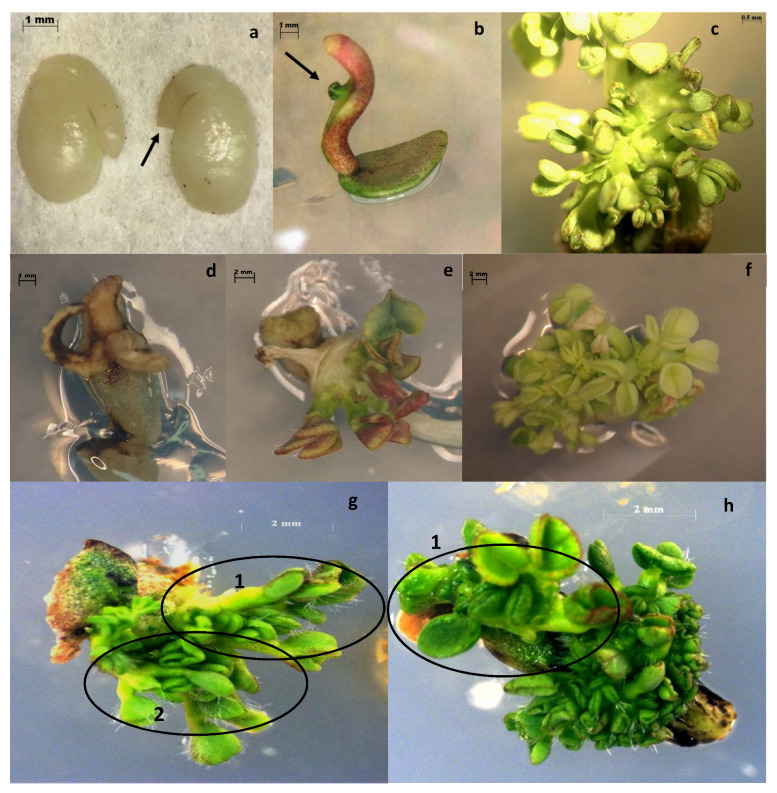
Seed cut procedure and regeneration: (**a**) seed cut with 1–2 mm segment of cotyledonary petiole attached (arrow showing cut), Scale = 1 mm; (**b**) shoot inducing in RM73 medium after 1 week (arrow showing shooting starting point), Scale = 1 mm; (**c**) multiple shoot generation through direct organogenesis after two weeks of culture. Dead explants identified at three stages; Scale = 0.5 mm; (**d**) Shoot apical meristem tissue was regenerated after 3 days in RM73 medium, but after two days in kanamycin 100 mg L^−1^, all explant tissue died; Scale = 1 mm; (**e**) Explant dying after 15 days of culture. The browning started at the top of the explant shoot; Scale = 2 mm; (**f**) Explant dying after 30 days of culture. The whole explant shows general tissue damage (yellowing); it was confirmed as dead one week after this photograph was taken; Scale = 2 mm; (**g**) Explant shoots were regenerated from the original apical meristem tissue. Two independent shoots can be distinguished; Scale = 2 mm (**h**) One shoot is clearly developed, whereas other plant tissue remains less differentiated; Scale = 2 mm. Images (**a**–**e**) were captured using microscope STEMI 2000-C (Carl Zeiss^®^, Oberkochen, Germany). Images g and h were captured with AxioCam Color MRm (Carl Zeiss^®^, Germany), software AxioVision V.4.5.

**Table 1 ijms-22-04181-t001:** Summary of the antibiotic sensitivity tests conducted with different concentrations. ^a^ Significant at the 1% level between treatments and control (proportion test).

		Percentage of Survival		
Treatment	Number of Explants	15 Days	30 Days	45 Days
Kanamycin concentration test				
0 mg L^−1^ (Control)	45	95.5	86.6	82.2
25 mg L^−1^	27	96.4	66.6	48.1
50 mg L^−1^	38	76.3	60.5	44.7
100 mg L^−1^	30	42.8 ^a^	23.3 ^a^	16.6 ^a^
Hygromycin concentration test				
0 mg L^−1^ (Control)	30	100	96.6	93.3
10 mg L^−1^	30	90	56.4 ^a^	36.6 ^a^
25 mg L^−1^	30	46.6 ^a^	30 ^a^	16.6 ^a^
50 mg L^−1^	30	16.6 ^a^	10 ^a^	0 ^a^
Cefotaxime concentration test				
0 mg L^−1^ (Control)	36	81.8	72.7	NA
10 mg L^−1^	33	86.1	80.5	NA
25 mg L^−1^	31	80.6	74.3	NA
50 mg L^−1^	29	82.7	79.3	NA

**Table 2 ijms-22-04181-t002:** Summary of the antibiotic combined tests conducted with different concentrations. ^a^ Significant difference at the 1% level between **a** and **b** populations. ^b^ No significant difference within **a** nor **b** populations (proportion test).

Treatment	Number of Explants	Percentage of Survival after 30 Days
Control (no antibiotics added)	29	89.6 ^a^
200 mg L^−1^ Cef	30	80 ^a^
100 mg L^−1^ Kan	30	13.3 ^b^
100 mg L^−1^ Kan + 200 mg L^−1^ Cef	29	27.5 ^b^
40 mg L^−1^ Hyg + 200 mg L^−1^ Cef	45	15.5 ^b^

**Table 3 ijms-22-04181-t003:** Summary of the *GUS* transformation efficiency statistics. Total of 379 explants tested: 5.2% of those were confirmed by PCR at explant level after 30 days.

Experiment	Total Number of Plants	No. Resistant Explants after 30 Days in Selection Medium (% in Parentheses)	PCR Configuration (% in Parentheses)
1	128	23 (17.9%)	9 (7.0%)
2	126	22 (17.4%)	7 (5.5%)
3	125	26 (20.8%)	4 (3.2%)

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
