# Peer review of "An Improved Protocol for Agrobacterium-Mediated Transformation in Subterranean Clover (Trifolium subterraneum L.)"

_ijms, 2021, doi:10.3390/ijms22084181_

Round 1
Reviewer 1 Report
The manuscript presents an improved protocol for the Agrobacterium-mediated transformation in the subterranean clover, which is one of the most important pasture legume crops in Australia. The subterranean clover offers many advantages: it is highly tolerant to grazing; it is very nutritious for livestock; and provides nitrogen fixation and soil structure improvement. However, it requires high soil phosphorus availability to overcome poor assimilation. The Agrobacterium-mediated transformation in subterranean clover provides a great tool for genetic improvement: improve soil phosphorus exploration; increase herbicide tolerance; increase the feeding value and many more. In this work, the authors improve the transformation efficiency by meticulously investigating many factors affecting the outcome of the transformation such as in vitro culture conditions, antibiotic sensitivity, Agrobacterium strain, shoot regeneration.
Despite the agricultural importance of subterranean clover, there is only a few protocols for the transformation and cell culture of this species.
The manuscript is very well written and all the steps in the protocol are well described. It is easy to follow and clear. The major point is that I cannot see any supplementary figures authors refer to. Below I listed some of the minor points for the authors and formatting issues for the production team.
Line 109:
Move Table 2 to the next page. It would look better if the Tables were spaced with the text.
Line 115:
I do not see the Figure S3.
Line 116:
I do not see the Figure S4.
Line 129
I do not see the Figure S5.
Line 134:
Figure S5?
Line 151
“Table 3. Summary of the GUS transformation efficiency statistics”
Move to another page.
Line 156
“1 %”
No space between 1 and %?
Line 161:
“The geocarpic habit leads to a high microorganism load on the seed coat in the species. totally contamination-free explants were not achieved in this study”
Change first period to comma or totally to Totally.
Line 186:
What does “escapes explants” mean?
Line 236:
Please make sure that the resolution of the Figure 1 is good enough.
The flowchart is very clear and easy to follow. Authors could add more of such schematic illustrations in the manuscript.
Line 261:
Figure 2: Are the different images of right proportions? They seem to be a bit deformed, especially Figure 2 g.
It would be nice to standardize the size of the scales and use the same font size. Images d, e and f miss the scale completely.
Letters could also be standardized regarding their position; a white box could also be used.
Make sure the image resolution is sufficient. The images look a bit blurred and overexposed.
Line 226:
In the Materials and Methods section, I can see a lot of details regarding the reagents used in the protocol. It would be nice to list all of them (with the number and provider) in a table.
Line 284:
“The experiment was repeated twice with kanamycin. The same experiment was performed for Hygromycin, testing 0, 10, 25, and 50 mg L-1 of this antibiotic and repeated twice.”
Hygromycin should be hygromycin.
Line 320:
Sigma
Is it Sigma-Aldrich?
Line 381:
I can see a list of Figures and Tables, which are not present in the text. Are those supplemental figures and tables? I could not see any supplement. Were these contents removed from the paper?
Author Response
Response to Reviewer 1 Comments
The manuscript presents an improved protocol for the Agrobacterium-mediated transformation in the subterranean clover, which is one of the most important pasture legume crops in Australia. The subterranean clover offers many advantages: it is highly tolerant to grazing; it is very nutritious for livestock; and provides nitrogen fixation and soil structure improvement. However, it requires high soil phosphorus availability to overcome poor assimilation. The Agrobacterium-mediated transformation in subterranean clover provides a great tool for genetic improvement: improve soil phosphorus exploration; increase herbicide tolerance; increase the feeding value and many more. In this work, the authors improve the transformation efficiency by meticulously investigating many factors affecting the outcome of the transformation such as in vitro culture conditions, antibiotic sensitivity, Agrobacterium strain, shoot regeneration.
Despite the agricultural importance of subterranean clover, there is only a few protocols for the transformation and cell culture of this species.
The manuscript is very well written and all the steps in the protocol are well described. It is easy to follow and clear. The major point is that I cannot see any supplementary figures authors refer to. Below I listed some of the minor points for the authors and formatting issues for the production team.
Line 109:
Move Table 2 to the next page. It would look better if the Tables were spaced with the text.
We have amended the document and the text moved down to space the tables. Table 2 is now in the next page.
Line 115:
I do not see the Figure S3.
We apologise for the oversight. Supplementary figures are included with the revised version.
Line 116:
I do not see the Figure S4.
Sorry. All supplementary figures are included in the revised version.
Line 129
I do not see the Figure S5.
As above - Supplementary figures are included within the revised version.
Line 134:
Figure S5?
As above - All supplementary figures are included with the revised version.
Line 151
“Table 3. Summary of the GUS transformation efficiency statistics”
Move to another page.
Moved.
Line 156
“1 %”
No space between 1 and %?
Corrected.
Line 161:
“The geocarpic habit leads to a high microorganism load on the seed coat in the species. totally contamination-free explants were not achieved in this study”
Change first period to comma or totally to Totally.
Adjusted in new Line 172-5 ‘The geocarpic habit leads to a high microorganism load on the seed coat in the species, this being the main cause of contaminations and consequent death of explants. Totally contamination-free explants were not achieved in this study.’
Line 186:
What does “escape explants” mean?
Clarification on text between brackets has been added as (explants that survive antibiotic stress).
Line 236:
Please make sure that the resolution of the Figure 1 is good enough.
We apologise for the poor-quality figure provided in the submitted manuscript. We have now uploaded figures with much improved quality in the revised version.
The flowchart is very clear and easy to follow. Authors could add more of such schematic illustrations in the manuscript.
We have now added more supplementary figures as suggested.
Line 261:
Figure 2: Are the different images of right proportions? They seem to be a bit deformed, especially Figure 2 g.
We apologise for the poor-quality figure provided in the submitted manuscript. We have now uploaded figures with improved quality.
It would be nice to standardize the size of the scales and use the same font size.
As images were taken with different cameras and software, unfortunately we are unable to standardize the scales. In the figure description we differentiate the equipment used for each image.
Images d, e and f miss the scale completely.
We apologise for this oversight. We have now uploaded figures with scales and further descriptions.
Letters could also be standardized regarding their position; a white box could also be used.
Letters standardized to Calibri.
Make sure the image resolution is sufficient. The images look a bit blurred and overexposed.
We apologise for the poor-quality figure provided in the manuscript. We have now uploaded figures with much improved quality.
Line 226:
In the Materials and Methods section, I can see a lot of details regarding the reagents used in the protocol. It would be nice to list all of them (with the number and provider) in a table.
Thanks for your suggestion. A table has now been added as a Supplementary Table (Table S1)
Line 284:
“The experiment was repeated twice with kanamycin. The same experiment was performed for Hygromycin, testing 0, 10, 25, and 50 mg L-1 of this antibiotic and repeated twice.”
Hygromycin should be hygromycin.
Changed to hygromycin.
Line 320:
Sigma
Is it Sigma-Aldrich?
This has been corrected to Sigma-Aldrich throughout.
Line 381:
I can see a list of Figures and Tables, which are not present in the text. Are those supplemental figures and tables? I could not see any supplement. Were these contents removed from the paper?
We apologise for the oversight and all supplementary figures and tables are included with the revised version.

Reviewer 2 Report
The manuscript entitled “An improved protocol for Agrobacterium-mediated transformation in subterranean clover (Trifolium subterraneum L.)” by Rojo et al. given efforts to improve the method of transformation protocol for subterranean clover. The study is scientifically novel, and the experiments carried out for it are well understood. However, there are a few comments that must be addressed.
Major comment
Authors focused on standardization of cefotaxime concentrations only not on the alterations of plant hormones and their ratios. No standardization of Agrobacterium OD, time of infection, incubation of temperature, the concentration of acetosyringone, and co-cultivation period which is not justifying the title.
Minor comments
- The abstract is simplified, and the authors primarily focused on antibiotic concentrations, so they must write details about plant hormones in their findings.
- What is RM73 media provide the composition in brief?
- mg L-1 rewrite this as mg L-1 these superscripts are highly ignored throughout the manuscript.
- Figure 3 may be given in the supplementary file and the rooting figure need to submit as the main figure.
- The OD of the Agrobacterium culture used in the present study is just 0.4 which is very less for general Agrobacterium transformation studies, have you standardized it? so need to provide clarifications/comparisons with previous findings
- According to table 2, the explant survival rate at control condition is 89.6% and at 200 mg L-1 cef it is reduced to 80% but authors gave a statement in heading of 2.3, a significant increase observed between the controls, 200 mg L-1 and 500 mg L-1 cef need to cross-check.
- Why there is much difference between putative plants raised from selection medium and PCR confirmed plants. of course, at this stage, they may not call as explants check in the table.
- The subheading of 4.3 is not suitable, modify it.
- Provide the PCR gel image as a supplementary file.
- The flow chart of figure 1 looks like a repetition of running text.
- In table 1, there are mixed results at cef concentrations of 10, 25, and 50 mg L-1 (up and down) the reason is not discussed in the discussion part.
- The discussion part in the manuscript is poor, mainly focused on future goals and studies instead of present findings and their significance. In the heading cefotaxime positive effective…………in the beginning of the para it is told that the concentration of 200 and 500 mg L-1 increased the explant fresh weight by 2X and 3X, respectively. But at the end of the para, it is written as an antagonist effect, 500 mg L-1 cefotaxime tended to reduce the mean explant fresh weight in relation to 200 mg L-1 need to clarify it.
- What is the purpose of standardization of kanamycin concentrations because it is not the selectable/bacterial marker not even increasing the explant proliferation?
Hence the overall recommendation is to address the minor suggestions in the manuscript with moderate revisions that meet the required suggestions mentioned above.
Author Response
Response to Reviewer 2 Comments
The manuscript entitled “An improved protocol for Agrobacterium-mediated transformation in subterranean clover (Trifolium subterraneum L.)” by Rojo et al. given efforts to improve the method of transformation protocol for subterranean clover. The study is scientifically novel, and the experiments carried out for it are well understood. However, there are a few comments that must be addressed.
Major comment
Authors focused on standardization of cefotaxime concentrations only not on the alterations of plant hormones and their ratios. No standardization of Agrobacterium OD, time of infection, incubation of temperature, the concentration of acetosyringone, and co-cultivation period which is not justifying the title.
Thank you for this comment. The current title is of ‘An improved protocol for transformation’ An improved protocol does not require alterations to all the factors involved in transformation e.g. plant hormones and their ratios, standardization of Agrobacterium OD, time of infection, incubation of temperature, the concentration of acetosyringone, and co-cultivation period etc. So, we consider the Title as valid. However, we have added text to both the Abstract and Discussion to reflect additional variables that could be tested by others.
See Abstract Lines 24-26: ‘Other variables additional to antibiotic and hormone combinations such as bacterial OD, time of infection and incubation temperature may be further tested to enhance even more the transformation.’
Also please see new Lines 239-243 in the Discussion: ‘Other variables may also be tested to enhance further the transformation efficiency, such as Agrobacterium OD and bacterial concentration (An OD=0.4, 24/36 as incubation time was used according to previous reports [10,11]), time of infection, incubation temperature, concentration of acetosyringone, and co-cultivation period time.’
Minor comments
- The abstract is simplified, and the authors primarily focused on antibiotic concentrations, so they must write details about plant hormones in their findings.
Information about hormones for rooting media was added to the abstract. E.g. Lines 19/20 ‘Different plant hormone combinations were tested to analyse the best rooting media. Roots were obtained in a medium supplemented with 1.2 µM IAA.’
- What is RM73 media provide the composition in brief?
Description of media composition in line 262, Materials and Methods, explant preparation and in vitro culture section. Reference 29.
- mg L-1 rewrite this as mg L-1 these superscripts are highly ignored throughout the manuscript.
Corrected.
- Figure 3 may be given in the supplementary file and the rooting figure need to submit as the main figure.
Figure 3 has been moved in the supplementary file (now Figure S6) and the rooting figure has been added as a main figure (now Figure 1).
- The OD of the Agrobacterium culture used in the present study was just 0.4 which is very less for general Agrobacterium transformation studies, have you standardized it? so need to provide clarifications/comparisons with previous findings.
Information about previous findings and reason of selection was added in the discussion section. See new Line 211/2 ‘(an OD=0.4, 24/36 as incubation time was used according to previous reports [10,11])’
- According to table 2, the explant survival rate at control condition is 89.6% and at 200 mg L-1 cef it is reduced to 80% but authors gave a statement in heading of 2.3, a significant increase observed between the controls, 200 mg L-1 and 500 mg L-1 cef need to cross-check.
No significant reduction in survival rate, but a significant increase in shoot regeneration and explant size. Heading 2.3 make references to figures S3 (explant size) and S4 (number of shoots per explant). Table 2 references the survival rate.
- Why there is much difference between putative plants raised from selection medium and PCR confirmed plants. of course, at this stage, they may not call as explants check in the table.
As there is a percentage of explants surviving the antibiotic selection (i.e. 15% of survival rate with 40 mg L-1 Hyg + 200 mg L-1 Cef ) it is expected to have some selected but untransformed explants (less positivity rate with PCR than number of resistant explants). This statement was added into the discussion section at Line 230/3
- The subheading of 4.3 is not suitable, modify it.
It is modified to Antibiotic concentration test
- Provide the PCR gel image as a supplementary file.
We apologise for the oversight and all supplementary figures are included with the reviewed version.
- The flow chart of figure 1 looks like a repetition of running text.
Thank you for your considered and insightful comments. We believe this makes it very easy to follow for readers as also appreciated by Reviewer 1 “The flowchart is very clear and easy to follow. Authors could add more of such schematic illustrations in the manuscript.”
- In table 1, there are mixed results at cef concentrations of 10, 25, and 50 mg L-1 (up and down) the reason is not discussed in the discussion part.
We have changed the Discussion in Lines 226-229 to ‘It was demonstrated that cefotaxime produced a positive impact in explant re-generation and did not significative affect the survival rate (Tables 1 and 2). Concentrations of 200/500 mg L-1 cefotaxime enhanced explant proliferation, increasing the number of shoots by 2X and their fresh weight by 3X, when compared with a negative control.’
The discussion part in the manuscript is poor, mainly focused on future goals and studies instead of present findings and their significance.
All the information requested was added in the discussion section. Additionally, more info about present findings was included as indicated above.
- In the heading cefotaxime positive effective…………in the beginning of the para it is told that the concentration of 200 and 500 mg L-1 increased the explant fresh weight by 2X and 3X, respectively. But at the end of the para, it is written as an antagonist effect, 500 mg L-1 cefotaxime tended to reduce the mean explant fresh weight in relation to 200 mg L-1 need to clarify it.
Regarding concentrations (200 and 500 mg L-1) - see above point- enhanced explant proliferation, increasing the number of shoots by 2X and their fresh weight by 3X, when compared with a negative control. But then, 500 mg L-1 showed a non-significant reduction in explant weight in comparison with 200 mg L-1. This latter information was clarified in the Discussion at Line 272 and thereafter.
- What is the purpose of standardization of kanamycin concentrations because it is not the selectable/bacterial marker not even increasing the explant proliferation?
Kanamycin, as well as hygromycin, is used as a selectable marker for transformation in a significant number of vectors. Even though we did not use it during our transformation protocol, we have optimised it concentration thinking ahead in the transformation with other vectors. See M. Bevan, Binary Agrobacterium vectors for plant transformation, Nucleic Acids Res. 12 (1984) 8711–8721.
Hence the overall recommendation is to address the minor suggestions in the manuscript with moderate revisions that meet the required suggestions mentioned above.

Reviewer 3 Report
Manuscript ID: ijms-1172838
Title: An improved protocol for Agrobacterium-mediated transformation in subterranean clover (Trifolium subterraneum L.)
The sustainable supply of foods relies directly on production. The ability to absorb nutrients from soil is crucial for plant growth and development. Authors addressed a significant limitation of Subterranean clover (Trifolium subterraneum L.), which is the poor capability to explore the soil P. Genetic manipulation can solve this issue, and Agrobacterium tumefaciens mediated gene transfer is a common protocol which is used in this study. The success rate of gene transfer quite impressive. However, there are several comments/questions as below that need to be answered for further improvement of the manuscript.
- In the abstract: methods need to define and write clearly in chronological order, followed by the results.
- In introduction: Why this study was conducted? The answer to this question is not clear. Authors need to addressed:
- Why has the subterranean clover cv. Daliak been chosen? Is this cultivar superior in terms of production, nutritional value etc.? What problems associated with this cultivar?
- What problems associated with the direct organogenesis of this species that would be cover come through optimizing culture conditions? What are the scopes of in vitro culture conditions optimization for this cultivar?
- Same for Agrobacterium tumefaciens mediated gene transfer. What are the factors that need to consider for the improvement of Agrobacterium tumefaciens transformation? As this article on the improved protocol of Agrobacterium tumefaciens transformation, without stating problems associated with the transformation and scope, readers will not get a clear view at the stage of background.
- In materials and Methods:
- How did the authors get the seeds of subterranean clover cv. Daliak? Is it purchased or collected from any germplasm? Mention it clearly.
- Based on the results in Table 1, the number of explants was different in different concentration of antibiotic, which is not justifiable. A clear justification of using the different number of explants required.
- Other sections of methods are well written, and statistical analyses were done appropriately. However, the acclimatization of plantlets information is absent. Without having this data, plant regeneration success is inconclusive.
- Results and discussion presentation are somewhat ok. Further discussion is required for the effect of f 0.3% chlorine and 0.8 mg mL-1 cefotaxime on explants. Indeed, chlorine and antibiotic will reduce the contamination, but what was the effect on the explants? Why the death of shoots occurred? The figure number is duplicated in the figure legend, and figure 1-6 (excluding figure 1-3 in materials and methods) is missing in the main manuscript. Therefore, further specific comments are not possible.
Author Response
Response to Reviewer 3 Comments
Title: An improved protocol for Agrobacterium-mediated transformation in subterranean clover (Trifolium subterraneum L.)
The sustainable supply of foods relies directly on production. The ability to absorb nutrients from soil is crucial for plant growth and development. Authors addressed a significant limitation of Subterranean clover (Trifolium subterraneum L.), which is the poor capability to explore the soil P. Genetic manipulation can solve this issue, and Agrobacterium tumefaciens mediated gene transfer is a common protocol which is used in this study. The success rate of gene transfer quite impressive. However, there are several comments/questions as below that need to be answered for further improvement of the manuscript.
- In the abstract: methods need to define and write clearly in chronological order, followed by the results.
Thank you for your considered and insightful comments. We have edited the abstract to reflect the suggestion with methods in chronological order, followed by the results.
- In introduction: Why this study was conducted? The answer to this question is not clear. Authors need to addressed:
- Why has the subterranean clover cv. Daliak been chosen? Is this cultivar superior in terms of production, nutritional value etc.? What problems associated with this cultivar?
The standard cultivar of subterranean clover cv. Daliak was chosen as its genome was fully sequenced (Text Ref 2). In that regard, so any modification in its genome after transformation can be analysed, knowing insertion loci, times inserted, genes affected etc.
Ref 2: Hirakawa, H.; Kaur, P.; Shirasawa, K.; Nichols, P.; Nagano, S.; Appels, R.; Erskine, W.; Isobe, S.N. Draft genome sequence of subterranean clover, a reference for genus Trifolium. Scientific Reports 2016, 6, 1-9.
- What problems associated with the direct organogenesis of this species that would be cover come through optimizing culture conditions? What are the scopes of in vitro culture conditions optimization for this cultivar?
Lines 52/3 are : “Only a few publications are available on the use of in vitro culture techniques to transform this species [9-11]. In all cases, direct organogenesis, using cut seeds as explants, was used to establish in vitro culture plants.
Same for Agrobacterium tumefaciens mediated gene transfer. What are the factors that need to consider for the improvement of Agrobacterium tumefaciens transformation? As this article on the improved protocol of Agrobacterium tumefaciens transformation, without stating problems associated with the transformation and scope, readers will not get a clear view at the stage of background.
We have added other variables that need to be considered for further optimization of Agrobacterium tumefaciens transformation (see above).
See from Line 240 on : “Other variables may also be tested to enhance further the transformation efficiency, such as Agrobacterium OD and bacterial concentration (An OD=0.4, 24/36 as incubation time was used according to previous reports [10,11]), time of infection, incubation temperature, concentration of acetosyringone, and co-cultivation period time.’
- In materials and Methods:
- How did the authors get the seeds of subterranean clover cv. Daliak? Is it purchased or collected from any germplasm? Mention it clearly.
Line 286 added: ‘(sourced from Dr Phillip Nichols of former Department of Agriculture and Food WA (DAFWA)’
- Based on the results in Table 1, the number of explants was different in different concentration of antibiotic, which is not justifiable. A clear justification of using the different number of explants required.
Added from Line 252: ‘Additionally, the process of seed coat removal, embryo tip excision, and cotyledonary petiole cut is quite difficult to perform, even for a qualified technician. As some seeds were lost in the process, the final number of explants in every treatment differs, but that was taken into consideration when the statistical analysis was performed.’
Other sections of methods are well written, and statistical analyses were done appropriately. However, the acclimatization of plantlets information is absent. Without having this data, plant regeneration success is inconclusive.
As we experienced problems with the rooting regeneration media, we could not present significant results at the plantlet level.
The section is edited from Line 245 on: “As transformation efficiency was tested at the explant level, it will be necessary to obtain a complete plant in further experimentation to test if the transgene is maintained through the generations in a Mendelian manner.”
- Results and discussion presentation are somewhat ok. Further discussion is required for the effect of f 0.3% chlorine and 0.8 mg mL-1 cefotaxime on explants. Indeed, chlorine and antibiotic will reduce the contamination, but what was the effect on the explants?
Line 63 onwards: ‘The seeds when grown according to the protocol of Ding et al. [10] showed contamination in 45.3 % of the cotyledons and in 57.7 % of embryo tips. But the addition of 0.3% chlorine and 0.8 mg mL-1 cefotaxime in the seed sterilization step reduced contamination - by comparison to 2%. In subsequent experiments, this percentage was maintained.’
Why the death of shoots occurred?
Mainly due to contamination. Subterranean clover differs from all other clovers in its underground seed development (geocarpy) [1]. The geocarpic habit leads to a high microorganism load on the seed coat in the species
Line 245 now reads: ‘The geocarpic habit leads to a high microorganism load on the seed coat in the species, this being the main cause of contaminations and consequent death of explants.’
The figure number is duplicated in the figure legend, and figure 1-6 (excluding figure 1-3 in materials and methods) is missing in the main manuscript. Therefore, further specific comments are not possible.
We apologise for the oversight, all the supplementary figures are included with the revised version.
